# Gene Regulation Network Analysis on Human Prostate Orthografts Highlights a Potential Role for the *JMJD6* Regulon in Clinical Prostate Cancer

**DOI:** 10.3390/cancers13092094

**Published:** 2021-04-26

**Authors:** Mario Cangiano, Magda Grudniewska, Mark J. Salji, Matti Nykter, Guido Jenster, Alfonso Urbanucci, Zoraide Granchi, Bart Janssen, Graham Hamilton, Hing Y. Leung, Inès J. Beumer

**Affiliations:** 1GenomeScan B.V. Plesmanlaan 1D, 2333 BZ Leiden, The Netherlands; 2352539C@student.gla.ac.uk (M.C.); m.grudniewska@genomescan.nl (M.G.); zoraide.granchi@gmail.com (Z.G.); b.janssen@genomescan.nl (B.J.); 2Institute of Cancer Sciences, University of Glasgow, Glasgow G61 1QH, UK; Mark.Salji@ggc.scot.nhs.uk; 3CRUK Beatson Institute, Glasgow G61 1BD, UK; 4Laboratory of Computational Biology, Institute of Biomedical Technology, Arvo Ylpön katu 34, 33520 Tampere, Finland; matti.nykter@uta.fi; 5Department of Urology, Erasmus Medical Center, Doctor Molewaterplein 40, 3015 GD Rotterdam, The Netherlands; g.jenster@erasmusmc.nl; 6Department of Tumor Biology, Institute for Cancer Research, Oslo University Hospital, 0424 Oslo, Norway; alfonsou@ifi.uio.no; 7Glasgow Polyomics, University of Glasgow, Glasgow G61 1QH, UK; Graham.Hamilton@glasgow.ac.uk

**Keywords:** prostate cancer, prognostic biomarkers, gene regulatory network, regulon, transcriptional regulator

## Abstract

**Simple Summary:**

Prostate cancer is a very common malignancy worldwide. Treatment resistant prostate cancer poses a big challenge to clinicians and is the second most common cause of premature death in men with cancer. Gene expression analysis has been performed on clinical tumours but to date none of the gene expression-based biomarkers for prostate cancer have been successfully integrated to into clinical practice to improve patient management and treatment choice. We applied a novel laboratory prostate cancer model to mimic clinical hormone responsive and resistant prostate cancer and tested whether a network of genes similarly regulated by transcription factors (gene products that control the expression of target genes) are associated with patient outcome. We identified regulons (networks of genes similarly regulated) from our preclinical prostate cancer models and further evaluated the top ranked *JMJD6* gene related regulated network in three independent clinical patient cohorts.

**Abstract:**

Background: Prostate cancer (PCa) is the second most common tumour diagnosed in men. Tumoral heterogeneity in PCa creates a significant challenge to develop robust prognostic markers and novel targets for therapy. An analysis of gene regulatory networks (GRNs) in PCa may provide insight into progressive PCa. Herein, we exploited a graph-based enrichment score to integrate data from GRNs identified in preclinical prostate orthografts and differentially expressed genes in clinical resected PCa. We identified active regulons (transcriptional regulators and their targeted genes) associated with PCa recurrence following radical prostatectomy. Methods: The expression of known transcription factors and co-factors was analysed in a panel of prostate orthografts (*n* = 18). We searched for genes (as part of individual GRNs) predicted to be regulated by the highest number of transcriptional factors. Using differentially expressed gene analysis (on a per sample basis) coupled with gene graph enrichment analysis, we identified candidate genes and associated GRNs in PCa within the UTA cohort, with the most enriched regulon being *JMJD6,* which was further validated in two additional cohorts, namely EMC and ICGC cohorts. Cox regression analysis was performed to evaluate the association of the *JMJD6* regulon activity with disease-free survival time in the three clinical cohorts as well as compared to three published prognostic gene signatures (TMCC11, BROMO-10 and HYPOXIA-28). Results: 1308 regulons were correlated to transcriptomic data from the three clinical prostatectomy cohorts. The *JMJD6* regulon was identified as the top enriched regulon in the UTA cohort and again validated in the EMC cohort as the top-ranking regulon. In both UTA and EMC cohorts, the *JMJD6* regulon was significantly associated with cancer recurrence. Active *JMJD6* regulon also correlated with disease recurrence in the ICGC cohort. Furthermore, Kaplan–Meier analysis confirmed shorter time to recurrence in patients with active *JMJD6* regulon for all three clinical cohorts (UTA, EMC and ICGC), which was not the case for three published prognostic gene signatures (TMCC11, BROMO-10 and HYPOXIA-28). In multivariate analysis, the *JMJD6* regulon status significantly predicted disease recurrence in the UTA and EMC, but not ICGC datasets, while none of the three published signatures significantly prognosticate for cancer recurrence. Conclusions: We have characterised gene regulatory networks from preclinical prostate orthografts and applied transcriptomic data from three clinical cohorts to evaluate the prognostic potential of the *JMJD6* regulon.

## 1. Background

Prostate cancer (PCa) is the second most common cancer among men and the fifth leading cause of death worldwide [1]. Tumour heterogeneity in PCa (between patients and among different tumour foci within individual patients) creates a major obstacle to the identification of clinically relevant molecular subtypes [2]. As a result, PCa treatment decisions are not based on tumour biology. Disease recurrence following treatment remains a significant problem, even following radical treatment such as radical prostatectomy or radical radiotherapy [3]. Despite the use of docetaxel chemotherapy or second generation androgen receptor pathway inhibitors along with androgen deprivation therapy (ADT), patients presenting with advanced and/or metastatic disease are at high risk of recurrent disease, which tend to be aggressive and incurable as either castration resistant (CRPC) or neuroendocrine PCa variants [4,5]. Therefore, there is an unmet need to improve our understanding of progressive PCa in order to identify new targets for therapy as well as prognostic biomarkers.

Inter-patient tumoral heterogeneity and intra-tumour heterogeneity among different tumour foci are well reported, making it unlikely that a single gene will be a representative biomarker of PCa progression [6]. Investigating a gene set-related network may leverage the correlations of the expression of multiple interacting genes [7]. Several gene set-based panels are offered as prognostic tests for PCa patients. Commercial assays [8,9,10] including Decipher™, Oncotype DX^®^ and Prolaris, together with scoring methods published in the literature, have been developed using microarray, Illumina or Nanostring transcriptome profiles [9,10,11] to apply mRNA expression data to predict the risk of cancer recurrence and/or progression. While gene expression-based models have resulted in promising data for predicting cancer behaviour in vitro [11], significant improvements are required before a stratification/prognostic tool in PCa patients can be considered for routine clinical practice, including the prediction of the risk of cancer recurrence following treatment [12]. The limitations of existing commercial molecular PCa diagnostic tests may stem from potential biases introduced during the signature identification step (including factors related to patient ethnicity [13], immune [14] and stromal [15] components of the tumours) that may influence the gene expression profiles. Moreover, gene set-based methods typically focus on the expression of individual genes or gene sets, without the ability to incorporate biologically important information associated with gene-gene interactions [7].

Alterations in transcriptional programmes are frequently implicated in PCa progression [16]. Genes that co-operate within the same biological pathways are often under the regulatory control of shared (one or more) transcription factors. Conveniently, interacting genes tend to be associated at the expression levels [17], providing the chance to infer their relationships from transcriptomics data. Gene regulatory networks (GRNs) are graphs describing transcriptional regulators and their target genes as nodes, while the relationships (level of correlation) among the regulators and target genes are presented as the edges. Statistical and/or machine learning approaches have been applied to gene expression data [18] to predict the topology of GRNs, namely the arrangement of transcriptional regulators and their target genes as well as the direction of each transcription factor-target interaction (i.e., positive or negative regulation). Within GRNs, data on the agreement between the predicted regulations and differential gene expression analysis can be applied to explore the underlying biological mechanisms to explain specific phenotypes (such as cancers with lower or higher chances of recurrence/progression).

Preclinical models of human PCa cells grown as orthotopic xenografts in mice (orthografts) represent a useful tool to mimic progressive clinical disease. However, the use of preclinical PCa as a tool to identify potential GRNs involved in progressive disease has not been tested. Here, to generate a robust scoring method, we derived GRNs from a collection of preclinical hormone naïve (dependent on androgens for growth) and castration resistant (growth despite androgen deprivation therapy) human PCa orthografts to capture the heterogeneous nature of clinical disease, leveraging the strength of correlations in the expression patterns of genes transcribed by tumour cells only. Filtering the GRNs for statistically significant associations led to the identification of putative regulons, signifying the network of target genes and shared transcription factor (or transcriptional regulator) involved. Integrating data from preclinical orthografts and clinical PCa cohorts, we modelled regulon signatures to identify patients at risk of cancer recurrence, and identified the *JMJD6* (Jumonji Domain Containing 6, arginine demethylase and lysine hydroxylase, a protein hydroxylase or histone demethylase) regulon as a prognostic marker in PCa (Figure 1).

## 2. Materials and Methods

See Supplementary Information for additional details for datasets and methods.

### 2.1. Datasets

Hormone naïve human prostate cancer cell lines (CWR, LNCaP and VCaP) were implanted into the prostates of androgen proficient (6 weeks old) nude male mice to generate androgen dependent prostate orthografts. Castration resistant (or androgen independent) prostate orthografts were generated from the 22Rv1, LNCaPAI and VCaPR human PCa cell lines by orthotopic implantation into the prostates of castrated nude (6 weeks old) male mice. RNA-seq data were obtained from 18 orthografts derived from the six human PCa cell lines studied (*n* = 3 mice per cell line) [19], referred to as the UGLA dataset. All data were included for the inference of the gene regulatory network.

RNA-seq data from three clinical PCa cohorts were included in this study: The University of Tampere (UTA-EGAD00001000609), the Erasmus Medical Center in Rotterdam (EMC-EGAD00001004215), and the International Cancer Genome Consortium (ICGC-EGAD00001004791). A summary of the clinicopathological characteristics of the cohorts is provided in Table 1.

Dataset from the UTA cohort [20] were obtained from 46 prostate tumour samples, including 28 untreated PCa samples from radical prostatectomy and 12 benign prostate hyperplasia control samples (obtained by radical prostatectomy, cystoprostatectomy or transurethral resection). RNA-seq data from treatment naive PCa samples that passed mapping quality control, provided with information on progression free time (*n* = 27), were used in this study, along with the 12 benign samples.

The EMC dataset was obtained from 92 radical prostatectomy specimens (51 PCa with 41 adjacent benign prostate tissue) [21,22]. The tumour content was confirmed histologically. Only prostate tumour samples with the information on progression free time (*n* = 37) and all the benign control samples were used in the present study. 

The ICGC dataset consists of 125 PCa specimens (and 8 matched benign control tissue) from 100 radical prostatectomy specimens [23]. Six tumour samples from the same prostatectomy specimens were sampled multiple times (from 3 to 6 biological replicates per patient) and were averaged at gene count level per patient, given the similarity in expression profiles. Samples from patients that did not receive neo-adjuvant therapy (*n* = 85) and all the benign samples (*n* = 8) were used in the present study.

### 2.2. Regulons Identification and Filtering

The PCa gene-regulatory network was generated using the R package ‘RTN’ [24] version v2.4.6, based on FPKM values (Fragments Per Kilobase of transcript per Million mapped reads) of the UGLA orthograft dataset and a list of 2065 transcription factors that were given as input (manually curated from MsigDb [25]). Out of the 2065 transcription factors, statistically significant associations with one or more target genes were found for 1643 regulators. The normalised counts matrix was then filtered by genes with FPKM equal or higher than one in at least one sample and standardised within the zero-to-one range. The function ‘tna.shadow’ from the R package ‘Viper’ [26] version 1.14.0 has been used to account for the ‘shadow’ effect (the chance of obtaining false positive result) during the enrichment of a GRN, if a non-active regulator shares a significant proportion of its targets with a bona fide active transcription factor, providing a final set of 1308 regulons. 

### 2.3. Gene Regulatory Network Metrics

The graph structure was analysed using the R package ‘igraph’ v1.2.5, exploiting the functions ‘degree’, ‘betweenness’, ‘constraint’ and ‘closeness’ to retrieve metrics at the ‘nodes’ level, providing complementary information about the importance of individual nodes within the network: (1) The ‘degree’ (or ‘in-degree’) of a node in a GRN is the number of transcriptional regulators involved in the control of the expression of a specific target gene. For different GRNs, the number of regulatory genes implicated for individual target genes varies, depending on complexity of the network; (2) ‘Betweenness’ is defined as the number of shortest paths passing through the node and can be interpreted as a measure of the influence of the node of interest over the global flow of information; (3) Burt’s ‘constraint’ is a measure of the redundancy of the information received by the node and can be interpreted as its ability to converge different signals; (4) ‘Closeness’ quantifies the node’s participation within a network. Finally, the Jaccard Index, a statistical measure defined as the ratio of the intersection and the union of two sets, was applied to highlight network nodes sharing a meaningful proportion of targets. The threshold of 0.1 was chosen to prioritise the nodes to be shown in Figure 2. A threshold of Jaccard Index/Co-efficient set at 0.1 highlights pairs of regulons with intersection (sharing) of ≥10% of the target genes when considered across the full set of target genes for the respective regulons.

### 2.4. Regulons Enrichment

The list of differentially expressed genes (DEG; ≥log2 fold changes and false discovery rate, FDR, ≤0.05) for each sample was determined using the respective benign control samples within each of the clinical UTA, EMC and ICGC cohorts. *p*-values were adjusted using the function ‘p.adjust()’ from the R stat package v4.0.3, by setting the ‘method’ parameter to ‘fdr’. The DEG gene set for each cohort was analysed using the gene regulatory network that was identified in the preclinical orthografts. The derived list of positive and negative gene-gene interactions was then used as input to the function ‘nbea’ from the package ‘EnrichmentBrowser’ [27] version v2.12.1, applying the ‘GGEA’ (gene graph enrichment analysis) method with default parameters. A threshold of FDR ≤0.05 has been adopted to identify the enriched regulons.

### 2.5. Statistical Analyses

Biochemical recurrence was defined as serum prostate-specific antigen (PSA) levels ≥2 ng/mL above nadir PSA (the lowest PSA level after treatment) and signifies clinical evidence of relapsed cancer. Relapse free survival (defined by absence of biochemical recurrence) was used to evaluate the prognostic utility of regulon signatures of interest in the UTA, EMC and ICGC clinical cohorts. The performance of our candidate *JMJD6* regulon signature as a prognostic marker was compared to three published signatures [28,29,30]. The performance of our candidate *JMJD6* regulon signature as a prognostic marker was compared to three published signatures (using the formulas used in the original publications [28,29,30]: (1) For the TMCC11 signature, the per-sample average of the normalised counts of the genes belonging to the signature was used to stratify the patient cohort into two groups according to values above or below the 67th percentile. (2) For the HYPOXIA-28 signature, the normalised counts were multiplied by the coefficient associated to each gene of the signature and all the products were added together to generate a sample-specific overall score, and the patient cohort was stratified into two groups according to the median of its distribution. (3) For the BROMO-10 signature, the function ‘gsva’ from GSVA v1.38.2 was used to analyse data from the normalised counts to calculate a signature enrichment score per sample. 

Patients were labelled according to the enrichment status of *JMJD6,* as predicted by GGEA, into active or inactive status groups. Hazard ratios (HR) for all the analysis were obtained by Cox proportional-hazard model regressions, using the ‘coxph’ function from the R package ‘survival’ version 3.1-8. Moreover, for multivariate analysis, Gleason score and the TNM (Tumour/Node/Metastasis) classification were added to the model formula in the form: ‘Endpoint ~ JMJD6regulon_activity + second_variable’. Kaplan–Meier curves were obtained using the ‘ggsurvplot’ function from the R package ‘survminer’ v0.4.8. The analysis was performed in R v4.03. 

## 3. Results

### 3.1. Gene Regulatory Network Inferred from Preclinical Prostate Orthograft Models

The expression profiles of 2064 manually curated transcription factors and co-factors [25] (Appendix A) were correlated with the differentially expressed genes in 18 prostate orthografts derived from human PCa cells, namely CWR22Res, 22Rv1, LNCaP, LNCaP-AI and VCaP (*n* = 3, except for VCaP). VCaP derived orthografts were grown in both hormone proficient and castrated mice (*n* = 3 each). 1308 regulons with a median of 20 genes per regulon (range 2–121) were identified (Appendix A). Interestingly, a large fraction of transcription factors (*n* = 607; 46.4%), shared at least one target gene (Figure 2A). 

Genes controlled by multiple transcription factors at the transcriptional level may suggest a functional requirement in controlling the expression of these target genes, thus signifying the likelihood of their biological importance. We searched for genes (as part of individual GRNs) predicted to be regulated by the highest number of transcriptional factors (Appendix A). Up to 10 transcription factors per target gene were observed within the networks identified. Four target genes were associated with the highest number of transcription factors (*n* = 10), and interestingly all of these four genes have previously been implicated in PCa: *BUD31* encodes for a bona-fide AR-coactivator that enhances AR transactivation in prostate cells [31]; *PLOD3* is involved in tissue remodelling and plays a role in multiple tumour types including PCa [32]; *SDR42E1* is implicated in early prostate organogenesis as well as carcinogenesis [33] and *XAGE1A* belongs to the cancer testis antigens family and its expression profile is linked to the aggressiveness of PCa [34]. Hence, a GRN-based analysis of prostate orthografts generated a network of candidate transcriptional regulators and their target genes that can be evaluated in clinical tumours.

### 3.2. Analysis of Differentially Expressed Genes (DEG) in Clinical PCa Patient Cohorts

Through comparison of each clinical tumour with the combined benign controls within the respective clinical cohorts, lists of differentially expressed genes (on a per sample basis) were generated on a per-sample basis initially in the UTA clinical cohort as part of a discovery analysis. The list of PCa associated genes (log2 fold changes and *p*-values) was then be used to identify the GRNs of interest, highlighting potential active regulons in individual tumours. In the UTA cohort (*n* = 27 PCa), we found a median of 2406 upregulated (range 1098–6419) and 282 downregulated (range 44–1173) genes per sample. In the EMC cohort as a validation dataset (*n* = 37 PCa), we observed a median of 2439 upregulated (range 827–7395) and 126 downregulated (range 1–925) genes for individual tumour samples. 

We ranked the differentially expressed genes by the average frequency of alteration (up- or down- regulation) within the respective patient cohorts (Appendix A). Of note, many of the frequently altered genes have been implicated in PCa, including *HPN* [35], *CLDN8* (an androgen regulated gene that promotes PCa cell proliferation and migration) [36], and *ONECUT2* (a known master regulator in PCa that suppresses the androgen axis) [37]. Hence, analysis of differentially expressed genes in the UTA and EMC cohorts highlighted candidate genes associated with PCa. 

### 3.3. Gene Graph Enrichment Analysis 

Data from transcription factor associated GRN identified in the preclinical prostate orthografts and individual gene sets from differentially expressed gene analysis on a per sample basis were integrated in a gene graph enrichment analysis (GGEA) to determine the activity status of the regulons (transcriptional regulators and their respective target genes) in the clinical tumours. The concordance of the positive and negative ‘transcription factor-target gene’ relationships was calculated for each sample within the UTA and EMC patient cohorts. GGEA [38] applies an enrichment approach to study the interactome surrounding the coregulators of interest to find supporting evidence of transcription factor activity. Differentially expressed genes in individual tumours within the two cohorts were mapped onto the candidate GRNs highlighted in the orthograft models.

To corroborate enriched gene networks shared among independent cohorts, we ranked the regulons by the respective frequency of activation in the UTA and EMC patient datasets (Appendix A). Consistently, among the ten most frequently active transcription factors (regulators) in these two datasets, we found three known genes implicated in PCa progression: *BACH1* promotes invasion and migration of PCa cells by altering metastasis related genes [39]; *CITED2* (Cbp/P300 Interacting Transactivator With Glu/Asp Rich Carboxy-Terminal Domain 2) has recently been proposed as a therapeutic target to tackle PCa metastasis [40]; and *DNMT1* promotes PCa metastasis through the regulation of epithelial-mesenchymal transition and cancer stem cells [41]. Collectively, regulatory patterns identified in our preclinical orthograft PCa models successfully highlighted genes of potential clinical relevance. 

### 3.4. Prognostic Utility of Regulon Activity Status in Radical Prostatectomy Clinical Cohorts

To evaluate the prognostic utility of the inferred regulons, we investigated the potential association between the enriched/not enriched status of regulons and the time to cancer relapse (signified by biochemical recurrence) following radical prostatectomy. We performed univariate CoxPH regression analysis in the UTA dataset in the first instance to identify enriched regulons associated with cancer recurrence (Appendix A). Eleven statistically significant candidate regulons highlighted, with *JMJD6* as the top-ranking enriched regulon (*p* = 0.002; Table 2A, Figure 2B, Appendix A). Analysing the EMC cohort as a validation dataset, fourteen enriched regulons were identified. Consistent with findings from the UTA cohort, *JMJD6* was also identified as the top-ranking enriched regulon (*p* = 0.003; Table 2B, Figure 2B, Appendix A). Besides *JMJD6*, the *SUFU* regulon was enriched in both UTA and EMC cohorts. Analysing all available prostate cancer datasets in the cBio-portal (*n* = 22), altered *JMJD6* gene was detected in multiple cohorts, with the highest incidence of genetic abnormalities (up to 8%) detected in metastatic tumours (Appendix A). We reasoned that analysis of the *JMJD6* regulon as a network, rather than at a single gene level, would provide additional insight into its functional impact. Univariate regression analysis further revealed that the active *JMJD6* regulon was associated with early biochemical recurrence in both UTA (discovery) and EMC cohorts (Table 3A). We further examined the status of the *JMJD6* regulon as a prognostic signature in the ICGC cohort for additional independent validation. Enrichment of the *JMJD6* regulon significantly correlated with time to biochemical recurrence in the ICGC cohort in univariate analysis (*p* = 0.00648). Kaplan-Meier analysis for biochemical free survival further confirmed reduced biochemical free survival in the presence of active status for the *JMJD6* regulon in patients within the UTA, EMC and ICGC cohorts (Figure 3).

To benchmark the *JMJD6* regulon as a prognostic marker for progressive/recurrent PCa, three reported independent signatures were selected for comparison: two androgen receptor related signatures (namely *TMEFF2* regulated cell cycle related gene signature [11] and the bromodomain related 10-genes signature [12]) as well as a 28-gene hypoxia signature [13]. The three signatures are referred to as TMCC11, BROMO-10 and HYPOXIA-28 respectively hereafter. Compared to *JMJD6* being prognostic in all three cohorts, TMCC11 was prognostic in the UTA and ICGC cohorts but not the EMC cohort, while BROMO-10 and HYPOXIA-28 significantly predicted recurrence in only one of the three cohorts, UTA and ICGC, respectively (Table 3A). Multivariate analyses of the three signatures (and of the *JMJD6* regulon status) were performed if the respective univariate analysis were significant. In multivariate analysis, the *JMJD6* regulon status significantly predicted disease recurrence in UTA and EMC, but not ICGC (Table 3B). Among the three published signatures, none significantly prognosticate for cancer recurrence in multivariate analysis. 

Collectively, our analysis highlights the feasibility of integrating data from preclinical human orthograft models of PCa with multiple clinical cohorts to generate information on the regulon landscape in identifying potential prognostic signatures. For the first time, our data identified the active status of the *JMJD6* regulon in patients at risk of PCa recurrence.

## 4. Discussion

We hypothesised that the study of genes positively and negatively regulated by one or more transcription factors (collectively referred to as regulons) is a suitable approach to capture the general mechanisms driving tumour progression in PCa [42]. For the first time, we integrated datasets from preclinical human prostate orthografts and clinical cohorts to investigate if specific regulons were associated with the outcome of patients with PCa. By mapping transcriptomic gene graph enrichment-based signatures on to a network of interacting gene regulators, we identify the *JMJD6* regulon as a candidate prognostic signature for biochemical recurrent PCa. Our data is consistent with a recent report on GRN-based investigation in breast cancer [43]. Our data on JMJD6 in PCa is consistent with involvement of JMJD6 in oral [44], breast [45], neuroblastomas [46], melanoma [47] and ovarian [48] cancers. 

The *JMJD6* regulon consists of 27 positive and 3 negative putative target genes (Appendix A), including *RAD51, EZH2* and *SORL1*. *RAD51* is predicted to be upregulated by *JMJD6* (Figure 2B). *RAD51*, a critical gene for the DNA repair process, is upregulated in aggressive PCa [49], and is included as part of the panel in the U.S. Food and Drug Administration approved Prolaris gene expression assay [50]. Similarly, *EZH2* (Enhancer of zeste homolog 2) is associated with PCa progression [51], and predicted to be upregulated by JMJD6 (Appendix A). Lastly, the expression of *SORL1*, a known hypoxia regulated gene [29], negatively correlates with JMJD6 expression.

We successfully identified regulons of interest from preclinical prostate orthografts and then investigated the prognostic value of our top candidate *JMJD6* regulon. Given the small number of preclinical samples available as a starting point to infer the GRNs in PCa, we were not able to robustly compare between hormone naïve and castration resistant orthografts. Instead, we combined the available orthografts to model tumour heterogeneity of clinical PCa. Importantly, some transcription factor-target genes relationships may not be revealed because of the limited sample number, thus creating potential biases with a subset of regulons appearing transcriptionally more important. Nonetheless, even with this limitation, the *JMJD6* regulon was identified as a key regulon enriched in two independent clinical cohorts, namely UTA and EMC, as well as the published independent ICGC clinical cohort. The ICGC cohort consists of relatively young patients (mean: 47, range: 35–52 years), compared to UTA (mean: 60, range: 47–71 years); such case selection bias may create confounding factors that contribute to the negative multivariate analysis for the *JMJD6* regulon in the ICGC cohort.

Although androgen receptor (AR) is essential for both prostate organogenesis and carcinogenesis, to our surprise, AR was not identified as an enriched regulon in our analysis. AR may be functionally important in both benign and malignant prostatic epithelium, with distinct transcriptional profiles arising from functional re-programming. Even in CRPC, AR remains activated through by-pass mechanisms despite suppressed canonical (classical) androgen receptor pathway activities [52]. In addition, changes due to reprogramming of the AR as a transcription factor may not be fully highlighted by analysis of regulons as fixed transcription factor-target genes ‘units’. Furthermore, AR splice variants (including AR-V7) are strongly implicated in CRPC. During the preparation of this report, a highly relevant publication highlighted the relationship between catalytic function of JMJD6 and the generation of AR-V7 mRNA in advanced prostate cancer [53]. Silencing of JMJD6 expression suppressed growth of LNCaP95 and 22Rv1 human CRPC cells, while combined *JMJD6* knockdown and anti-androgen treatment with enzalutamide produced substantially more anti-proliferative effects than each of the two treatment alone. Collectively, their data implicates JMJD6 to be important in PCa cell viability and proliferation, thus further supporting our GRN-based findings. 

The strategy of standardising the analysis, by adopting a panel of benign controls within each dataset (benign prostatic hyperplasia for the UTA and ICGC cohorts; benign tissue adjacent to the tumour for EMC cohort), allowed for the reduction of biases arising from different protocols for sample handling, sequencing and data processing. Indeed, by leveraging a panel of control samples within each cohort, it was possible to show commonalities among the independent data sets without resorting to batch correction. 

JMJD6 belongs to the Jumonji C (JMJC) domain-containing family of proteins. JMJD6 is thought to function mainly as a lysyl 5-hydroxylase, and not as a demethylase [54], although enzymatically it has been shown to possess both catalytic activities. Its ability to regulate the transcriptional activity of p53 through hydroxylation of a lysine in the p53 C-terminus is highly relevant in cancer biology. Upregulated JMJD6 expression is related to tumour growth, tumour metastasis and high tumour pathological classification [55,56,57]. To build on our findings, the classical Waddington epigenetic landscape [58] model can be applied to describe in more detail the mechanism of regulation for the target genes within the *JMJD6* regulon. Given its potential role in a number of tumour types, a novel JMJD6 specific inhibitor SKLB325 has recently been developed [48]. Should future research confirm *JMJD6* as a driver gene for progressive PCa, formal evaluation of JMJD6 targeted therapy will be warranted.

## 5. Conclusions

We have characterised gene regulatory networks from preclinical prostate orthografts and applied transcriptomic data from three clinical cohorts to identify the *JMJD6* regulon as a potential prognostic marker in PCa. 

## Figures and Tables

**Figure 1 cancers-13-02094-f001:**
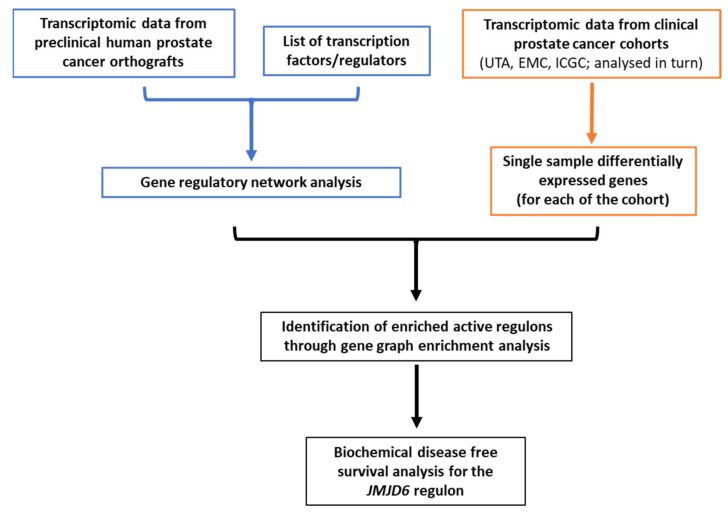
Workflow summarising the study analysis pipeline.

**Figure 2 cancers-13-02094-f002:**
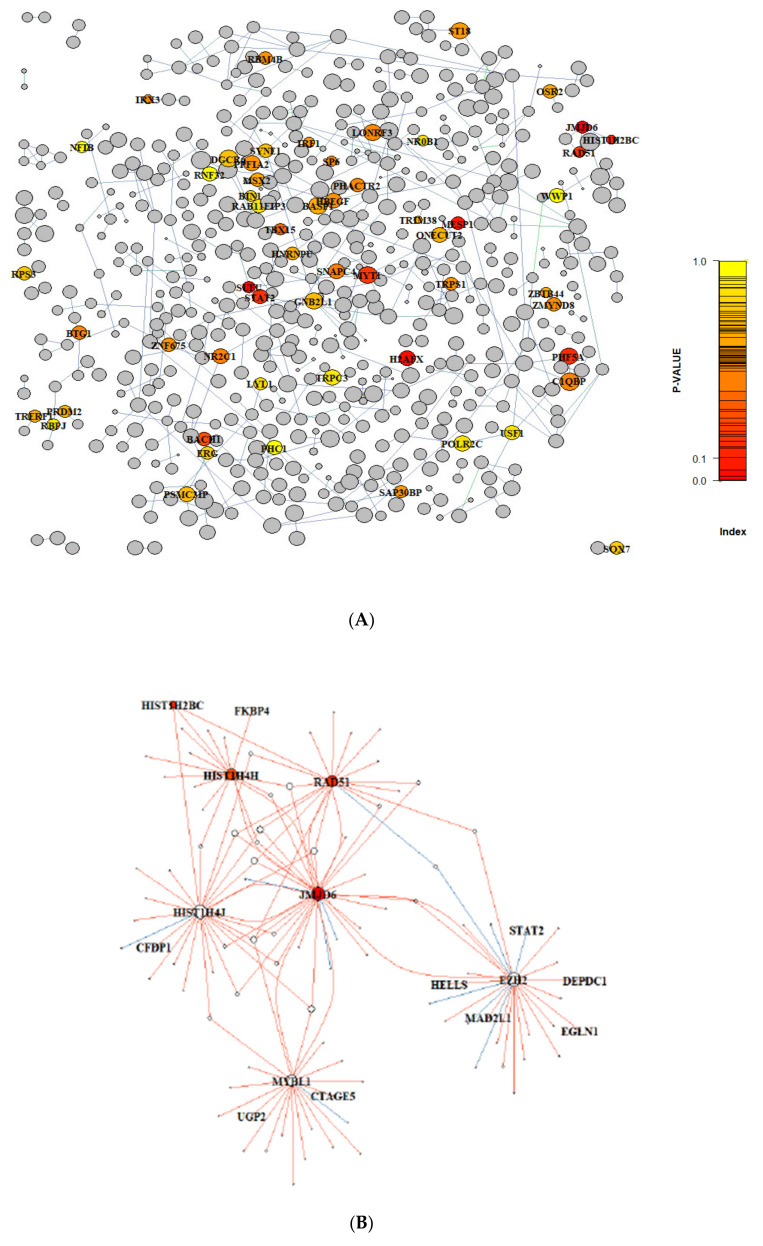
(**A**) Gene regulatory networks identified in preclinical human prostate cancer orthografts. The regulatory network of regulons (nodes of all colours) is presented with the edges linking pair of regulons sharing part of their targets. The commonality between pairs of regulons was calculated through the Jaccard index. Pairs with Jaccard Index ≥0.1 are shown. The colour of the nodes refers to the colour scale (range 1–0) represents the *p*-value of the enriched regulons associated with relapse free survival in the clinical (UTA and EMC) cohorts by cox regression analysis. Regulons in grey represent insignificant networks and therefore not included in further analysis. (**B**) The gene regulatory network topology cantered on the *JMJD6* regulon. Red edges represent positive regulations while blue edges inhibitory relationships. (**A**,**B**) The names of regulators are annotated with HUGO gene symbol in black. The colour scale (range 1–0) represents the *p*-value of the enriched regulons associated with disease free survival.

**Figure 3 cancers-13-02094-f003:**
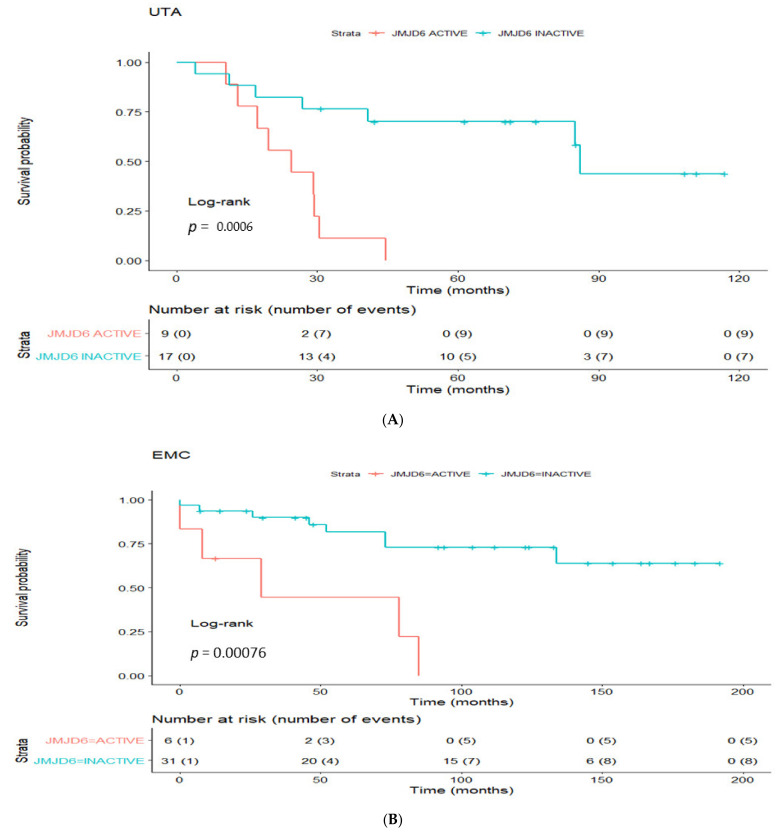
Disease free survival analysis of the *JMJD6* regulon signature in clinical prostatectomy patient cohorts. The survival probability curves for patients in the UTA (**A**), EMC (**B**) and IGCG (**C**) cohorts were prepared with patients stratified according to the presence or absence of the enriched *JMJD6* regulon in red and turquoise, respectively.

**Table 1 cancers-13-02094-t001:** Clinicopathological characteristics of patient cohorts (NA, data not available).

Clinical Cohorts	UTA		EMC		ICGC	
Number (*n*)	*n* = 27	%	*n* = 37	%	*n* = 85	%
age at diagnosis						
range	47–71		NA		32–52	
mean	60		NA		47	
median	61		NA		48	
na	0					
psa at diagnosis (ng/mL)						
range	3.5–48.1		0.3–36.2		3.1–743	
mean	10.4		11.8		30.48	
median	8.3		9.4		8.21	
na	0		0		0	
tumour stage						
t1	10	37.0	1	2.7	0	0.0
t2	16	59.3	15	40.5	61	71.8
t3	1	3.7	13	35.1	23	27.1
t4	0	0.0	8	21.6	1	1.2
na	0	0.0	0	0.0	0	0.0
gleason score						
<7	7	25.9	6	16.2	12	14.1
7	13	48.2	19	51.4	65	76.5
>7	7	25.9	0	0.0	8	9.4
na	0	0.0	12	32.4	0	0.0
therapy						
Radical prostatectomy	27	100	37	100	85	100

**Table 2 cancers-13-02094-t002:** Univariate cox regression analysis for regulons enrichment. Top ten genes are listed for the (**A**) UTA and (**B**) EMC cohorts.

(A)		
Ensembl ID	Hugo Symbol	*p* Value
ENSG00000070495	*JMJD6*	0.002
ENSG00000196132	*MYT1*	0.006
ENSG00000100410	*PHF5A*	0.02
ENSG00000065057	*NTHL1*	0.02
ENSG00000159210	*SNF8*	0.02
ENSG00000171222	*SCAND1*	0.02
ENSG00000123091	*RNF11*	0.02
ENSG00000120798	*NR2C1*	0.02
ENSG00000107882	*SUFU*	0.03
ENSG00000146083	*RNF44*	0.04
(**B**)		
**Ensembl ID**	**Hugo Symbol**	***p*** **Value**
ENSG00000070495	*JMJD6*	0.003
ENSG00000095002	*MSH2*	0.006
ENSG00000107882	*SUFU*	0.007
ENSG00000136826	*KLF4*	0.01
ENSG00000119969	*HELLS*	0.01
ENSG00000151929	*BAG3*	0.02
ENSG00000105607	*GCDH*	0.02
ENSG00000092607	*TBX15*	0.02
ENSG00000188486	*H2AFX*	0.02
ENSG00000180596	*HIST1H2BC*	0.03

**Table 3 cancers-13-02094-t003:** Cox regression univariate (**A**) and multivariate (**B**) survival analysis. *p*-values (P), Hazard ratios (HR) and 95% Confidence intervals (CI) are showed for each univariate regression. Multivariate analysis results, using Gleason score and/or tumour stage as covariates, are shown only for the variables whose association with biochemical recurrence was significant (*p* < 0.05) at univariate level. All significant *p*-values are highlighted in bold.

(A) Univariate Analysis									
Clinical Cohorts	UTA			EMC			ICGC		
Statistics	HR	95% CI	*p*	HR	95% CI	*p*	HR	95% CI	*p*
**Clinicopathological variables**									
Gleason score	2.7	1.6–4.7	**0.0004**	1.9	0.2–16	0.5	2	1.4–3	**0.0004**
Tumor stage	1.7	1–2.96	0.05	1.3	1–1.6	**0.02**	2.5	1.7–3.8	**<0.0001**
**Signatures**									
active *JMJD6* regulon	6	1.9–18	**0.002**	5.8	1.8–18.6	**0.003**	4.2	1.5–12	**0.006**
TMCC11	4.5	1.1–17.8	**0.03**	1	0.3–3.7	1.0	4	1.6–10.5	**0.004**
BROMO-10	0.06	0.0069–0.52	**0.01**	1.2	0.3–4.2	0.8	2.6	0.7–9.3	0.2
HYPOXIA-28	2.1	0.7–6.24	0.2	1.1	0.4–3.5	0.8	3.4	1.3–9.2	**0.01**
**(B) Multivariate analysis**									
	**UTA**			**EMC**			**ICGC**		
	HR	95% CI	P	HR	95% CI	P	HR	95% CI	P
**JMJD6 regulon**	6.5	1.3–32	**0.02**	4.4	1.3–14.6	**0.01**	1.2	0.3–4.8	0.7
Gleason score	1.6	0.8–3.1	0.2				1.2	0.6–2.4	0.6
Tumor stage	2.3	1.1–4.9	0.03	1.2	1–1.5	0.05	2.6	1.3–4.9	**0.004**
**TMCC11**	3.4	0.8–14.4	0.1				1.8	0.6–5.6	0.3
Gleason score	2.5	1.4–4.4	**0.002**				1.3	0.7–2.4	0.5
Tumor stage	1.58	0.8–3.2	0.2				2.2	1.1–4.4	**0.02**
**BROMO-10**	0.3	0.03–4.2	0.4						
Gleason score	2.15	1.08–4.27	0.03						
Tumor stage	1.5	0.8–2.8	0.2						
**HYPOXIA-28**							2.1	0.7–6.1	0.2
Gleason score							1.3	0.7–2.4	0.4
Tumor stage							2.2	1.13–4.24	**0.02**

## Data Availability

The data presented in this study are openly available. The orthograft sequencing data has been uploaded in array express at www.ebi.ac.uk/arrayexpress/ (accessed on 26 November 2020), with access details available from the corresponding authors. The codes used in the report are available at https://github.com/MarioCangiano/Cancers_article.git (accessed on 7 April 2021).

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
