# Peer review of "Gene Regulation Network Analysis on Human Prostate Orthografts Highlights a Potential Role for the JMJD6 Regulon in Clinical Prostate Cancer"

_cancers, 2021, doi:10.3390/cancers13092094_

Round 1
Reviewer 1 Report
Cangiano et al present a gene signature for predicting relapse. They begin from cell line derived orthographs and use these to identify sets of coregulated genes (regulons) which they predict to be regulated by the same transcription factor. They then compare the activity of these regulons between normal controls and cancer samples from two different clinical datasets and identify a regulon that is identified as “enriched” in both clinical datasets. They then use this regulon as a prognostic predictor in the training datasets and in a third independent validation dataset. While the regulon is predictive when in single regression in all three datasets, it is not predictive in a multivariate regression in the validation dataset. They compare to other published signatures and find some of the other signatures are prognostic in some of the datasets in single-regression, but no signature is prognostic in a multiple-regression.
The approach is basically sound, and the results are interesting. I have two key concerns with the paper that must be addressed, and a number of smaller concerns.
Firstly, the lack of significance in the multiple regression in the ICGC dataset is somewhat worrying as it means that evidence is lacking that this signature provides a benefit after accounting for tumour grade. Since the JMJD6 signature was discovered in the EMC and UTA datasets, we can really only measure its usefulness in the ICGC dataset, particularly when comparing to other signatures. The authors should at the very least discuss and acknowledge this shortcoming further. The only point of discussion is that perhaps validation is hampered in ICGC by selection bias. Can the authors support the usefulness of their signature further? What happens if they divide their UTA and EMC datasets into test and validation subsets? Does the JMJD6 dataset show up in such a dataset? Is it significantly predictive in a validation subset held out from the discovery process?
Secondly, the orthograph sequencing data used to discover the regulons is not available, either through a public repository linked in this work, nor the work referenced as the source of this data (from some of the same authors). This is completely unacceptable.
I further have a number of comments, concerns and suggestions:
- There is no code available to support the analyses done here.
- “Single sample differential expression” is poor terminology for the process described and suggests a type of statistics other than what is performed.
- The exact details of the single sample differential expression is unclear to me. The authors state that DEGs were identified by “subtracting the Deseq2 normalised counts of each tumoral sample with the average of all benign prostatic samples within each dataset.” Which type of normalisation is being done here (I would suggest rlog or vst)? Are the normalised counts on a log or a linear scale (it should definitely be on a log scale if the authors want to use Gaussian statistics)?
- In the section “Fastq pre-processing” it is unclear what the source of the annotations is. The authors state “GRCh37.75”, but this seems like a genome version, rather than an annotation version.
- It is unclear to me how the scores for other signatures have been calculated? Did they use the same GGCA based enrichment approach? Did each signature come with its own score calculation method?
- If other signatures use a more traditional scoring method, is the difference in performance between the authors signature and the others, the gene content or the signature, or the scoring method? Could they apply the GGCA approach to the other signatures? Or could they apply a more traditional scoring method to their signature?
- Lines 90-91: The authors state that GRNs have regulators as nodes and relationships between regulators and targets as edges. Shouldn’t targets also be described as nodes in this definition?
- Throughout the methods, there is a lack of citation of Bioconductor packages. The authors need to include citations to at the least RTN, Viper and EnrichmentBrowser.
- Line 149: The authors refer to 1,643 TFs. It was unclear to me where this number came from for some time, as it sounds like only 1,643 TFs were used for the analysis (rather than RTN only found regulons for that number, which I believe to be the meaning?)
- Line 161: Is there any evidence to support the supposition that tightness of regulation is correlated with node degree?
- Figure 2: It is unclear to me what the node colour represents. I am also unclear what the phrase “with the edges proportional to the Jaccard index of each pair of regulons” means.
- Line 189: Please describe how p-values were turned into FDRs for this analysis
Author Response
Reviewer 1
Cangiano et al present a gene signature for predicting relapse. They begin from cell line derived orthographs and use these to identify sets of coregulated genes (regulons) which they predict to be regulated by the same transcription factor. They then compare the activity of these regulons between normal controls and cancer samples from two different clinical datasets and identify a regulon that is identified as “enriched” in both clinical datasets. They then use this regulon as a prognostic predictor in the training datasets and in a third independent validation dataset. While the regulon is predictive when in single regression in all three datasets, it is not predictive in a multivariate regression in the validation dataset. They compare to other published signatures and find some of the other signatures are prognostic in some of the datasets in single-regression, but no signature is prognostic in a multiple-regression.
The approach is basically sound, and the results are interesting. I have two key concerns with the paper that must be addressed, and a number of smaller concerns.
Firstly, the lack of significance in the multiple regression in the ICGC dataset is somewhat worrying as it means that evidence is lacking that this signature provides a benefit after accounting for tumour grade. Since the JMJD6 signature was discovered in the EMC and UTA datasets, we can really only measure its usefulness in the ICGC dataset, particularly when comparing to other signatures. The authors should at the very least discuss and acknowledge this shortcoming further. The only point of discussion is that perhaps validation is hampered in ICGC by selection bias. Can the authors support the usefulness of their signature further? What happens if they divide their UTA and EMC datasets into test and validation subsets? Does the JMJD6 dataset show up in such a dataset? Is it significantly predictive in a validation subset held out from the discovery process?
Response: We thank Reviewer 1 for the suggestion in using of the UTA and EMC sequentially with one as discovery and the other as a first step towards validation. We entirely agree with this suggestion. This was in fact one of the options in our original plan for the structure of the manuscript. The UTA cohort was studied first and is therefore presented as the discovery cohort in the revised manuscript. The EMC was performed afterwards, and is now considered as a validation cohort. We have now edited the manuscript accordingly.
Secondly, the orthograph sequencing data used to discover the regulons is not available, either through a public repository linked in this work, nor the work referenced as the source of this data (from some of the same authors). This is completely unacceptable.
Response: We appreciate Reviewer’s comment about availability of the sequencing data. The orthograft sequencing data has been uploaded in array express at www.ebi.ac.uk/arrayexpress/, with access deatails: E-MTAB-9831 Username: Reviewer_E-MTAB-9831; Password: nvspmxvg
I further have a number of comments, concerns and suggestions:
- There is no code available to support the analyses done here.
Response: We appreciate this comment. In fact, we have the code prepared but not sure whether this journal requires for it to be included. We have now included the code used in publicly available repository at https://github.com/MarioCangiano/Cancers_article.git.
- “Single sample differential expression” is poor terminology for the process described and suggests a type of statistics other than what is performed.
Response: We thank the reviewer for this comment. We have now revised the manuscript and rephrase the term ‘single sample differential expression’ to differentially expressed genes for each sample when compared to that observed in benign prostate samples (control) within the same cohort.
- The exact details of the single sample differential expression is unclear to me. The authors state that DEGs were identified by “subtracting the Deseq2 normalised counts of each tumoral sample with the average of all benign prostatic samples within each dataset.” Which type of normalisation is being done here (I would suggest rlog or vst)? Are the normalised counts on a log or a linear scale (it should definitely be on a log scale if the authors want to use Gaussian statistics)?
Response: The information can be found in the Supplementary Information. We have further updated this section to include more details:
Differential expression analysis
For each of the clinical cohorts (namely UTA, EMC and ICGC respectively), the fold changes of each differentially expressed gene (DEG) were calculated on a per-sample basis from the ratio of Deseq2 normalised counts for the tumoral sample and the average Deseq2 normalised counts of the control benign prostatic samples within the dataset, and is expressed in a log2 scale. The significance of the ratio was calculated through the interpolation on the standardised Gaussian distribution, after dividing each counts difference (Deseq2 norm counts of tumour minus Deseq2 norm counts of the normal group) by the standard deviation of the Deseq2 norm counts in the panel of normal samples. Deseq2 normalised counts were extracted with the Deseq2 function ‘counts()’ by setting the ‘normalized’ parameter to ‘True’. P-values were adjusted using the function ‘p.adjust()’ from the R stat package v4.0.3, by setting the ‘method’ parameter to ‘fdr’. For the analysis of DEGs for each sample, a threshold of FDR was set at adjusted p-value ≤0.05 and log2 fold change above or below zero have been used to identify genes up or down-regulated, respectively.
- In the section “Fastq pre-processing” it is unclear what the source of the annotations is. The authors state “GRCh37.75”, but this seems like a genome version, rather than an annotation version.
Response: The file we referred to is from the webpage: http://ftp.ensembl.org/pub/release-75/gtf/homo_sapiens/, and the file name is Homo_sapiens.GRCh37.75.gtf.gz
- It is unclear to me how the scores for other signatures have been calculated? Did they use the same GGCA based enrichment approach? Did each signature come with its own score calculation method?
- If other signatures use a more traditional scoring method, is the difference in performance between the authors signature and the others, the gene content or the signature, or the scoring method? Could they apply the GGCA approach to the other signatures? Or could they apply a more traditional scoring method to their signature?
Response: We thank Reviewer 1 for this insightful question. Additional details are included in the methodology section of the revised manuscript, and are highlighted below.
“The performance of our candidate JMJD6 regulon signature as a prognostic marker was compared to three published signatures (using the formulas used in the original publi-cations [28–30]: (1) For the TMCC11 signature, the per-sample average of the normal-ised counts of the genes belonging to the signature was used to stratify the patient co-hort into two groups according to values above or below the 67th percentile. (2) For the HYPOXIA-28 signature, the normalised counts were multiplied by the coefficient associated to each gene of the signature and all the products were added together to generate a sample-specific overall score , and the patient cohort was stratified into two groups according to the median of its distribution. (3) For the BROMO-10 signature, the function ‘gsva’ from GSVA v1.38.2 was used to analyse data from the normalised counts to calculate a signature enrichment score per sample.”
In summary, we employed a novel gene regulatory network approach to nominate regulon implicated by the transcriptome of a panel of human orthografts and then further investigate the candidate JMJD6 regulon, with additional benchmarking analysis using three published signatures. For future studies, it would be interesting to apply the respective approach to the various signature to probe their relationship of the signatures and the ‘connectivity’ of the different approaches.
- Lines 90-91: The authors state that GRNs have regulators as nodes and relationships between regulators and targets as edges. Shouldn’t targets also be described as nodes in this definition?
Response: We thank the Reviewer for this comment. We have updated the manuscript to clarify this point. Regulators and targets are nodes while the relationships (level of correlation) among them are referred to as the edges.
- Throughout the methods, there is a lack of citation of Bioconductor packages. The authors need to include citations to at the least RTN, Viper and EnrichmentBrowser.
- Line 149: The authors refer to 1,643 TFs. It was unclear to me where this number came from for some time, as it sounds like only 1,643 TFs were used for the analysis (rather than RTN only found regulons for that number, which I believe to be the meaning?)
Response: We have updated the manuscript to provide more details and clarity for the above two points.
The PCa gene-regulatory network was generated using the R package ‘RTN’[24] version v2.4.6, based on FPKM values (Fragments Per Kilobase of transcript per Million mapped reads) of the UGLA orthograft dataset and a list of 2,065 transcription factors that were given as input (manually curated from MsigDb [25]). Out of the 2,065 transcription factors, statistically significant associations with one or more target genes were found for 1,643 regulators. The normalised counts matrix was then filtered by genes with FPKM equal or higher than one in at least one sample and standardised within the zero-to-one range. The function ‘tna.shadow’ from the R package ‘Viper’ [26] version 1.14.0 has been used to account for the ‘shadow’ effect (the chance of obtaining false positive result) during the enrichment of a GRN, if a non-active regulator shares a significant proportion of its targets with a bona fide active transcription factor, providing a final set of 1308 regulons.
- Line 161: Is there any evidence to support the supposition that tightness of regulation is correlated with node degree?
Response: We have revised the text to clarify the point raised by Reviewer. In the text we are referring to ‘in-degree of the node within the GRN.
“The ‘degree’ (or ‘in-degree’) of a node in a GRN is the number of transcriptional regulators involved in the control of the expression of a specific target gene. For different GRNs, the number of regulatory genes implicated for individual target genes varies, depending on complexity of the network.”
- Figure 2: It is unclear to me what the node colour represents. I am also unclear what the phrase “with the edges proportional to the Jaccard index of each pair of regulons” means.
Response: We appreciate that the phrase “with the edges proportional to the Jaccard index of each pair of regulons” for Figure 2 is not easy to follow. We have removed this phrase and added the following description in the legend of Figure 2.
The regulatory network of regulons (nodes of all colours) is presented with the edges linking pair of regulons sharing part of their targets. The commonality between pairs of regulons was calculated through the Jaccard index. Pairs with Jaccard Index ≥0.1 are shown. The colour of the nodes refers to the colour scale (range 1-0) represents the p-value of the enriched regulons associated with relapse free survival in the clinical (UTA and EMC) cohorts. Regulons in grey represent insignificant networks and therefore not included in further analysis.
- Line 189: Please describe how p-values were turned into FDRs for this analysis
Response: P-values were adjusted using the function ‘p.adjust()’ from the R stat package v4.0.3, by setting the ‘method’ parameter to ‘fdr’.

Reviewer 2 Report
Figure 3 A&B lack units for the “Time” axes. The axis title should be “Time (days)”. Fig. 3C should have an axis title of “Time (months)”. The x-axes for A,B, & C use three different units for time. This is confusing for the reader. The data should be presented with a common set of time units so that the probability plots can be directly compared. This is particularly important because the patients in the EMC protocol appeared to fair much worse than the others. This is not explained in the text.
Table 2. It is simply ridiculous to quote P values to more than one, or at most two significant figures. The fact that the P values for PHF5A – RNF11 are all the same leads me to suspect that the number of genes or networks found was identical and might perhaps be an artefact? The same is true for GCDH and TBX15. Also, there are only two genes in common in A with B.
Table 3. See previous comment about P values. Usually P values are quoted to only one significant figure. Adding more digits does nothing to enhance the probability.
Overall: The growth, harvesting and processing of the orthografts needs to be presented. How long did the tumors grow in the mice? Did they all grow at the same rate? How large were they when they were harvested? How did the investigators avoid necrotic regions in some of the xenografts? Is it possible that these human tumor cells changed in response to growth in a mouse and that these changes formed a common alteration, rather than reflecting endogenous properties of the cell lines. One could easily test this by examining the expression data for the input cell types, but this control was not performed. Without this crucial control, I don’t think that these data should be published.
Author Response
Reviewer 2
Figure 3 A&B lack units for the “Time” axes. The axis title should be “Time (days)”. Fig. 3C should have an axis title of “Time (months)”. The x-axes for A,B, & C use three different units for time. This is confusing for the reader. The data should be presented with a common set of time units so that the probability plots can be directly compared. This is particularly important because the patients in the EMC protocol appeared to fair much worse than the others. This is not explained in the text.
Response: We thank Reviewer 2 for pointing out this omission. We have updated the label for x-axis for Figure 3, ensuring all three panels are consistent.
Table 2. It is simply ridiculous to quote P values to more than one, or at most two significant figures. The fact that the P values for PHF5A – RNF11 are all the same leads me to suspect that the number of genes or networks found was identical and might perhaps be an artefact? The same is true for GCDH and TBX15. Also, there are only two genes in common in A with B.
Response: We appreciate Reviewer 2’s comment about the P values presented in Table 2. We have revised the manuscript and data presented to ensure they are correct and to one significant figure.
Table 3. See previous comment about P values. Usually P values are quoted to only one significant figure. Adding more digits does nothing to enhance the probability.
Response: We agree with Reviewer 2’s point and have updated Table 3 accordingly.
Overall: The growth, harvesting and processing of the orthografts needs to be presented. How long did the tumors grow in the mice? Did they all grow at the same rate? How large were they when they were harvested? How did the investigators avoid necrotic regions in some of the xenografts? Is it possible that these human tumor cells changed in response to growth in a mouse and that these changes formed a common alteration, rather than reflecting endogenous properties of the cell lines. One could easily test this by examining the expression data for the input cell types, but this control was not performed. Without this crucial control, I don’t think that these data should be published.
Response: Reviewer 2 is right to consider details regarding the orthografts mentioned in the manuscript. Details of the orthografts are the subject of previous publications (Salji et al, 2021 published as a pre-print: http://dx.doi.org/10.2139/ssrn.3762111). To ensure that the readers of our report can readily obtain information on these details, we have updated the methodology section to explain the details as part of the supplementary information, as below.
Human PCa cell lines were authenticated using the Promega GenePrint 10 System. CWR - RRID:CVCL_LI38, 22RV1 - RRID:CVCL_1045, LNCAP-RRID:CVCL_4783, LNCAPAI -RRID:CVCL_4791 and VCAP - RRID:CVCL_2235. Hormone naïve cell lines CWR LNCAP and VCAP were maintained in RPMI 1640 with 2 mmol Glutamine and 10% Foetal Bovine Serum (FBS). Charcoal Stripped FBS (CSS) was used for maintenance of CR cell lines 22RV1 and LNCAPAI. 14×106 PC cells in serum free RPMI were mixed with matrigel (1:1), with final volume of 50 μl, and orthotopically injected into the anterior prostate of 10 week old male CD-1 Nude mice (Charles River Labs) +/- surgical castration (Project Licence P5EE22AEE), reviewed by local ethics committee in full compliance with UK Home Office regulations (UK Animals (Scientific Procedures) Act 1986).
In vivo experiments were performed in accordance with the ARRIVE guidelines, and were reviewed by a local ethics committee under the Project Licence P5EE22AEE in full compliance with the UK Home Office regulations (UK Animals (Scientific Procedures) Act 1986). Prostate cancer cells were suspended in serum-free RPMI medium and mixed 1:1 with Matrigel (Corning, NY, USA). Briefly, 14×106 cells (in 50 μl) were injected into the anterior prostate of CD1-nude mice (Charles River Laboratories, Wilmington, MA, USA). For CRPC conditions, orchidectomy was performed at the time of injection. Tumour growth was monitored weekly using A Vevo3100 ultrasound imaging system (Fujifilm Visualsonics, The Netherlands). Tumours were then allowed to grow for 9 weeks before reaching endpoint. At the end of the experiment, tumour orthografts were collected and weighted. Half of the tumour material was fixed in 10% formalin for histological procedures and the other half was snap-frozen in liquid nitrogen for transcriptomic analysis.
Reviewer 2 raised the issue of tumour heterogeneity within the orthografts. This is a highly relevant area of our research interest, particularly following treatment such as castration (or androgen deprivation therapy). We have recently published our work on the evidence of tumour heterogeneity following castration as indicated by functional imaging by positron emission imaging with correlation to molecular markers at mRNA and protein levels (EJNMMI Res 2020 Nov 25;10(1):143. doi: 10.1186/s13550-020-00728-9). We would suggest that these considerations are outside the remit of the current manuscript.

Reviewer 3 Report
The authors did some interesting work about basic science on prostate cancer cells that were implanted in mice to generate castration and non-castration-resistant PCa cell lines. They tested different genes of the regulatory network from preclinical PCa orthografts and applied transcriptomic data from three clinical cohorts to evaluate the prognostic potential of the JMJD6 regulon.
JMJD6 regulon seems to be a promising biomarker but still far from a clinical application. Furthermore, the use of this definition of BCR (definition normally used after radiotherapy) limits the reproducibility in a real cohort of patients undergoing radical prostatectomy. Try to use a different definition and see how it performs.
Furthermore, the BCR as an outcome is questionable due to the lack of a real parameter that affects the survival
Replace reference 1 : [1] Siegel RL, Miller KD, Fuchs HE, Jemal A. Cancer Statistics, 2021. CA Cancer J Clin 2021;71:7–33. https://doi.org/10.3322/caac.21654.
Author Response
Reviewer 3
The authors did some interesting work about basic science on prostate cancer cells that were implanted in mice to generate castration and non-castration-resistant PCa cell lines. They tested different genes of the regulatory network from preclinical PCa orthografts and applied transcriptomic data from three clinical cohorts to evaluate the prognostic potential of the JMJD6 regulon.
JMJD6 regulon seems to be a promising biomarker but still far from a clinical application. Furthermore, the use of this definition of BCR (definition normally used after radiotherapy) limits the reproducibility in a real cohort of patients undergoing radical prostatectomy. Try to use a different definition and see how it performs.
Response: We thank the positive comments from Reviewer 3. Taking into consideration with the comment about BCR, we have revised the manuscript using the term ‘Relapse Free Survival’ to indicate the patient status.
Furthermore, the BCR as an outcome is questionable due to the lack of a real parameter that affects the survival
Response: We agree with Reviewer 3 that we wish to ensure that the impact of our report is as wide as possible. In fact, the cohort that we analysed were on patients who underwent radical prostatectomy. With the revised terminology with ‘relapse free survival’, the readers will be able to interpret our findings accordingly. In the literature, disease free survival or relapse free survival is associated with overall survival in patients with prostate cancer (J Clin Oncol. 2009 Jun 10; 27(17): 2766–2771. doi: 10.1200/JCO.2008.18.9159).
Replace reference 1 : [1] Siegel RL, Miller KD, Fuchs HE, Jemal A. Cancer Statistics, 2021. CA Cancer J Clin 2021;71:7–33. https://doi.org/10.3322/caac.21654.
Response: We are unsure the issue with this reference, and have checked the information contained within this reference and can confirm that the citation is correct.

Round 2
Reviewer 2 Report
The authors have adequately addressed my concerns.